# “*If You Don’t See the Dog, What Can You Do?”* Using Procedures to Negotiate the Risk of Dog Bites in Occupational Contexts

**DOI:** 10.3390/ijerph18147377

**Published:** 2021-07-10

**Authors:** Sara C. Owczarczak-Garstecka, Robert M. Christley, Francine Watkins, Huadong Yang, Carri Westgarth

**Affiliations:** 1Department of Livestock and One Health, Institute of Infection and Global Health, University of Liverpool, 8 West Derby Street, Liverpool L69 7BE, UK; Robc@liverpool.ac.uk; 2Institute for Risk and Uncertainty, University of Liverpool, Chadwick Building, Peach Street, Liverpool L7 7BD, UK; 3Dogs Trust, Canine Behaviour and Research Team, 17 Wakley Street, London EC1V 7RQ, UK; 4Public Health, Policy & Systems, Institute of Population Health, University of Liverpool, Waterhouse Building, Block B, Brownlow Street, Liverpool L69 3GL, UK; Fwatkins@liverpool.ac.uk; 5Management School, University of Liverpool, Chatham Street, Liverpool L69 7ZH, UK; huadong.yang@liverpool.ac.uk

**Keywords:** dog bites, interviews, risk management, safety procedures, qualitative methods, workplace safety

## Abstract

Dog bites are a health risk in a number of workplaces such as the delivery, veterinary and dog rescue sectors. This study aimed to explore how workers negotiate the risk of dog bites in daily interactions with dogs and the role of procedures in workplace safety. Participants who encounter dogs at work were recruited using snowball sampling. Ethnographic methods (interviews, focus group discussions, participant-observations) were used for data collection. All data were coded qualitatively into themes. Six themes describing dog bite risk management were identified: ‘Surveillance of dogs’; ‘Communicating risk; ‘Actions taken to manage perceived risk’; ‘Reporting bites and near-misses’, ‘Investigating bites and near-misses’, and; ‘Learning and teaching safety’. While the procedures described dog bite risk as objective, when interacting with dogs, participants drew on experiential knowledge and subjective judgment of risk. There was a discrepancy between risks that the procedures aimed to guard against and the risk participants were experiencing in the course of work. This often led to disregarding procedures. Paradoxically, procedures generated risks to individual wellbeing and sometimes employment, by contributing to blaming employees for bites. Dog bite prevention could be improved by clarifying definitions of bites, involving at risk staff in procedure development, and avoiding blaming the victim for the incident.

## 1. Introduction 

Although it is recognised that the presence of dogs in society can be beneficial for human health and wellbeing [1] there are also risks to navigate. The incidence of dog bites differs between countries (e.g., 12.39 per 100,000 in Australia [2], 1.5 per 100,000 in the Netherlands [3], 25.3–30.1 per 100,000 in India [4], and 110 per 100,000 the US [5]). The incidence of dog bite-related hospitalisations in the UK is increasing and in 2018, the mean annual incidence reached 14.99 per 100,000 population [6]. The incidence rates were also correlated with the indices of multiple deprivation and unevenly distributed across the UK, meaning that the rate in some areas was much higher (e.g., 24.2 per 100,000 in Knowsley, North-West of the UK) and lower in others (e.g., 1.1 per 100,000 in City of London, London) [6]. In the UK, a population-based cross-sectional survey estimated that 25% of respondents in the UK have been bitten by a dog during their lifetime, however only a third of these bites required medical attention [7]. Nonetheless, between March 2014 and February 2015 alone, dog bites led to 7227 hospital admissions in England and Wales [8] and in 2017/2018 the cost of dog bites to the NHS was estimated at £70.8 millions [6].

Dogs are the second most commonly implicated species of animals (after insects) in all animal-related non-fatal injuries within the US workforce [9]. Among those injured, the most affected occupations are: non-farm animal caretakers, truck drivers, veterinary technicians and meter readers [9]. Moreover, in two separate studies carried out in Brazil and Taiwan, approximately 70% of surveyed postal workers reported being bitten by a dog at some point during their career [10,11]. On average 277 people are seriously bitten each year at work in the UK, making dog bites an important issue concerning workplace safety [12]. Indeed, dog bites are the second most common cause of injury for UK postal workers (after slips and falls), with an average of 7 postal workers being bitten each working day [13]. Furthermore, 48% of 2800 surveyed Australian veterinarians were bitten by a dog during the previous 12 months [14] and in the USA 63% of surveyed veterinarians have been bitten during the course of their career [15]. In the UK, two out of three veterinarians were injured at work at some point and 78% of these injuries were animal bites, including dog bites [16]. Dog bites are also common among dog shelter workers and other professionals whose work requires entering private properties [12]. 

Most countries develop national policies aimed to prevent dog bites. These typically involve defining a dog bite as a punishable offence, often in conjunction with breed specific legislations (BSL) that restrict the list of dogs that can be legally owned on grounds of safety. Although studies investigating the prevalence of dog bites concluded that BSL or similar legislations was successful in Winnipeg (Canada) [17], it was not linked with a reduction in a number of dog-related hospital admissions in The Netherlands [3], Ireland [18,19], Spain [20], Denmark [21] or England [22]. In addition, as banned breeds are usually not accurately identified [23,24,25,26], these legislations contribute to poor welfare of dogs perceived as dangerous [27,28,29]. Little is however known about organisation-wide policies aimed at bite prevention. 

Previous research on bite-prevention in the occupational context highlighted that organisations often rely on procedures for bite prevention [12]. Procedures (i.e., an established series of actions conducted in a certain order or manner [30], and their formalised, written version-protocols, are an intrinsic part of work practices, including those focused on risk management. However, procedures reflect “work as imagined”, i.e., how management perceive work practices and are therefore what is reflected in guidelines compiled in formal documents. This may be different to “work as actually done”- the everyday risk-work involved in identifying and managing risk, which could combine the formal protocols and personal, experience-based routines [30,31]. Protocols are developed on the basis of understanding risk as an objective phenomenon that estimates the severity and likelihood of an event and then maps directly onto the underlying hazard, i.e., anything that can cause harm [32]. In practice, the hazard caused by a dog and the risk around it may not be perceived in this way. Individual, subjective experiences of dogs could shape what one identifies as a hazard. Consequently, understanding of risk will also differ between individuals and shape their practices. 

Further, unlike technical contexts where procedures are used to manage risk not encountered in daily life (e.g., nuclear industry), dogs are common and encounters with dogs are frequent and often unexpected. Most people have some experience with dogs outside of the workplace, which may also contribute to their interactions with dogs as well as their take on workplace procedures as they may approach dogs in a routine manner. Routines, defined as a course of actions followed regularly and developed organically through practice, can also contribute to safety and may overlap with procedures. In addition, routines have also been described as a source of comfort, because they help to dispel any uncertainty regarding the best way of managing risk in daily lives, making them convenient to adhere to [33].

Therefore, management of dog bites can offer insights into interactions between different understandings of risk and different modes of risk management more generally. This tension has practical implications; whilst discrepancy between the formal rules and actual practice is normal, too large a gap can pose a considerable challenge to the management of employee safety [34]. The interplay between “work as imagined” and “work as actually done” is therefore particularly interesting in the context of managing risk a round dogs, and understanding the difference can contribute to improving dog bite prevention.

### Managing Risk through Procedures 

Work safety procedures are assumed to help to manage risk by minimising the variation of human behaviour and thus reducing the chance of human error [35]. It has been argued that the proliferation of procedures to manage risks, reflects a broader change in work-place environments [36,37] and changes in perceptions of risk [38]. The drive to improve work efficiency has resulted in an individual worker being more likely to work on a specific aspect of a task rather than see it through from beginning to an end [36]. These changes require work to be prescriptive, standardised and therefore inscribed in procedures [37,39]. This manner of standardising work often means that work outputs and processes become quantifiable, which enables and promotes understanding risk as an objective, measurable phenomena [38].

Safety protocols can lead to safer practices. For example, reduction in incidents in aviation and medicine has been attributed to an introduction of checklists [40]. However, provision of procedures does not guarantee an improvement in incident prevention [36]. Procedures can also shift the responsibility for safety management from the organisation to frontline staff, as it is easy to identify when their behaviour was not compliant with the procedure (regardless of whether it actually contributed to an incident, or was consistent with the demands of the task in hand). Therefore, a by-product of safety procedures can be “blame-the-worker” attitude and result in a change in power relations between the workers and employer [41]. In addition to managing the risk of injury, organisational safety procedures can maintain existing organisational culture, knowledge, values and norms [42]. For instance, organisations may use accident prevention protocols to manage their reputations and to shape their public image [41].

This paper reports part of a larger project exploring perception and prevention of dog bites. Here, the role of work procedures specifically is explored in depth. Given the lack of understanding of what people do to stay safe around dogs, the first aim of this paper is to explore the role of work procedures in management of risk around dogs, as it emerged as an important theme in previous research [12]. The second aim was to discuss different ways of understanding risk and safety around dogs which emerge when procedures are enacted in practice. The final aim was to provide recommendations for dog bite prevention. 

## 2. Materials and Methods

### 2.1. Theoretical Approach

A qualitative methodology was adopted in this study. Specifically, we used ethnographic methods which are defined broadly as “an approach to experiencing, interpreting and representing culture and society” [43] (p.18). The purpose of ethnographic research is to identify shared patterns of ideas, beliefs, language, actions, practices, understandings in reference to the cultural context of an individual or a group and to describe them using thick and rich descriptions [44]. Ethnographic methodology does not follow any prescriptive methods, however an immersion in the culture of research participants through fieldwork and participant-observations (see Section 2.2 and 2.3 for more details) is seen as important [44]. Ethnographic methods take a flexible, iterative and responsive approach to the study findings. For example, observations made during the fieldwork can be used to modify questions asked during the in-depth interviews and the preliminary interview findings can help to focus the participant-observations [44]. The goal of using an ethnographic approach here was to examine people’s experiences and practices within naturalistic settings, whilst paying attention to the broader socio-cultural and environmental contexts within which these were situated [45]. To this end, a multi-sited ethnographic approach was used to explore the meaning of dog related risk at multiple fieldwork sites and in different contexts [46].

Symbolic interactionism theory was used to interpret the data. Meaning (of an action, word, interaction between people, etc.) was therefore seen as acquired in the course of interactions and interpretations of these interactions [47] and not as inherent to the object. For example, in this study the meaning of “dangerous”, “nasty” or “vicious” dogs as well as “risk” was not assumed, but gauged from observing interactions [31]. 

### 2.2. Description of Study Sites 

Research took place between July 2016- July 2018, within the North-West of England, the area with the highest prevalence of dog bites in the UK [8]. The primary fieldwork sites were selected to enable access to employees of dog shelters and delivery companies, i.e., among the most frequently bitten occupations in the UK [12]. Fieldwork was conducted in two different branches of two delivery companies and in three different shelters run by two different organisations. The sites within the Delivery Companies and Dog Shelters differed with respect to their location (urban or rural) and number of employees (large and small). Dog shelters also differed regarding kennel design, equipment used and veterinary support. 

### 2.3. Data Collection 

To develop an in-depth understanding of the culture of dog bite prevention within work contexts, data were collected using a range of ethnographic methods: participant-observations, in-depth semi-structured interviews, focus-group discussions and analysis of documents related to fieldwork sites 

Participant-observations lasted 12 weeks and involved observing and joining staff in all aspects of their day-to-day work in delivery companies and dog shelters. After individual observations, a focus group discussion was run at each of the three shelters, attracting 7, 6 and 4 participants, respectively. Focus group discussions were conducted to explore how participants compare their experiences and practices with colleagues [44] and to enable study participation, as many members of staff did not had time for individual interviews. Insights from participant-observations were used to shape the questions during the interviews and focus-group discussions, facilitating research triangulation ([48]; Table 1). In addition, policy and training documents pertaining to preventing dog bites were collected and included in the analysis 

All interviews and focus-group discussions were performed in a quiet, private space selected by participants and lasted between 31 and 197 min. 

### 2.4. Participant Recruitment

Within each workplace, contact was first made with the organisation’s gatekeepers to facilitate the initial introductions to members of staff, advertise the study within the organisation and arrange the first interview. Later, a snowball sampling method was used to recruit participants [44]. Participants were recruited based on their experience of bites (i.e., being bitten or avoiding bites at work), duration of employment, role within the organisation, gender and age, so that a variety of participants and contexts could be included in the study. Recruitment stopped when no new themes were emerging from analysis [49]. 

### 2.5. Data Analysis 

Data analysis was carried out iteratively and concurrently with data collection to inform the ongoing research [45]. Detailed fieldwork notes including observations of participants, researcher’s reflections and memos (ideas about codes, themes and links with the existing literature), were kept throughout the process [49]. To reflect the role of the researcher in knowledge construction (i.e., the influence of positionality), fieldwork notes were considered during the analysis [49]. 

Interview and focus-group discussion data were transcribed to enable analysis. Thematic analysis was used to analyse all collected data (including policy and training documents collected during fieldwork and notes from participant-observations) and was used iteratively. This analysis included multiple steps, starting with thorough familiarisation with the collected data, followed by two coding cycles [49]. During both cycles, codes were developed quasi-inductively. By this we mean that codes were developed to best summarise the content whilst being relevant to the research objectives and theoretical literature on risk, which helped to narrow down the coding framework [49]. During the first coding cycle, descriptive codes were developed. During the second cycle, the descriptive codes were revised and replaced with analytical codes. Codes were discussed between the co-authors (S.C.O-G, C-W, F.W, R.C, and C.W) to improve rigour and consistency. Analytical codes were then grouped together on the basis of their similarity, which is how the main themes were identified [49]. The first coding cycle was conducted on paper; the second cycle relied on qualitative coding software NVIVO [50]. 

### 2.6. Ethical Considerations

This study was approved by the University of Liverpool Research Ethics Committee (project number: 1497). All participants were informed about the purpose of the research and gave written consent before focus-group discussions and interviews and verbal consent before observations. Decision-makers from all organisations granted written permission to carry out the study within their premises. Data were anonymised during transcription and when making notes by removing identifiable characteristics of interviewees, locations, incidents and dogs.

## 3. Results

### 3.1. Participants Characteristics

A total of 55 participants took part in this study (36.4% identified as men and 63.6 as women). Sixteen participants (50.0% men and 50.0% women) were interviewed and 39 (30.8% men and 60.2% women) were joined during participant-observations across all fieldwork sites. The average age of participants was 39 and the age range was 19–55. Participants’ experiences ranged from not being bitten to being bitten 3 times, and from bites that did not puncture skin to digit amputations.

### 3.2. Thematic Analysis 

Themes reflected different sets of procedures and routines of dog bite prevention: surveillance of dogs, communicating risk actions taken to manage perceived risk, reporting bites and near-misses, investigating bites and near-misses and learning and teaching safety (Figure 1). Procedures are represented here as distinct, in reality however they were often interconnected and overlapping, for example surveillance of dogs led to changing a person’s behaviour as they were reacting to what they saw. 

#### 3.2.1. Surveillance of Dogs– Identifying Hazards

Defining hazards: A crucial part of procedures for surveillance in delivery companies and dog shelters was identifying what should be noticed, i.e., what is a hazard. Although, hazards were not always explicitly defined in dog shelters, surveillance was centred on noticing dog body language. All participants from dog shelters watched for the same behaviours, e.g., changes in posture, position of a tail, ears, lip licking, raising paws or head turning. At the same time, interpreting dogs’ body language also relied on subjective judgement:

*“He’ll lift his paw, he lies on his back, and it’s just him being him (…) I know that I can still touch him when he’s doing that (…) but if anybody else went in that kennel, they’d probably be like, “Shit, I’m not going anywhere near this dog,” because he’s showing every sign that he doesn’t want you to go and attach the lead”*.Annie, Dog Shelter.

Surveillance protocols: In dog shelters, formal surveillance protocols were based on learning about a dog’s past behaviours from owners through a relinquishment forms and structured interview, observations of specific dog behaviours throughout their stay in the shelter and formal assessments. For instance, one of the formal assessments aimed to provide a measurable understanding of hazard posed by the dog was based on a score attributed to a number of behaviour observations. The list of behaviours to identify was based on understanding some dog behaviours as objectively hazardous. The subjective aspect of the assessment was minimised by relying on specifically trained staff and corroborating scores between assessors. At the same time, the protocol was headed with a question: *“Always think to yourself when filling in this assessment would you consider living with this dog?* This introduced the need to rely on subjective judgement to enact the hazard identification protocol.

Efficacy of surveillance protocols was limited when staff had to interact with dogs before they were formally assessed, when dog behaviour changed during their stay in a shelter and when behaviours were difficult to observe or interpret: 


*“The worse dogs are those that give no affection – you know, a dog that’s not being funny with you, but also not being nice. “What do you think!? Just give me something!””*
Alice, Dog Shelter.

While in shelters surveillance procedures were focused on observing dog behaviour, in delivery companies they were based on scanning the environment for presence of dogs. When dogs were not observed, delivery workers were asked to rattle the gates to make dog appear in order to avoid them:


*“When you’re [delivering mail or parcels] (…) you risk assess in your mind every property that you go to. You’re listening out for any slight noise. You’re looking for any indication that there’s a dog. (…)”*
Ben, Delivery Company.

One of the delivery companies explicitly defined “dog hazards” as:


*“Any animal that poses a threat while attempting to deliver or collect mail [or parcels]. Including:*


*Any dog or animal which: has attacked previously, (…) is restrained to avoid contact with delivery staff due to the likelihood (…) attack, (…) is roaming in its garden/ territory that presents a risk of attack (…), that shows aggression to other dogs or human, (…) that the delivery staff are uncomfortable with, (…) dogs behind letter boxes snapping at letters during delivery (…)”*.Delivery Company.

Objective and subjective surveillance: Being able to list all hazards suggests that hazards were objective. However, hazard identification also required recognising when a dog made one *feel* uncomfortable. Similar to dog shelters, enacting the safety protocols in the delivery company required drawing on a subjective judgement, which meant that *any* encountered dog could be perceived as a hazard. 

Failed surveillance: Within delivery companies, surveillance procedures failed when there were no signs of a dog living on the property, when it did not appear when gates were rattled, or when they escaped through open doors. Schematic representation of surveillance of dogs is summarised in Figure 2.

#### 3.2.2. Communicating Risk to Others

Communication protocols: Shelters and delivery companies aimed to make the risk of dogs visible to other workers. In shelters, risk was communicated by displaying information about the dog in multiple places (e.g., on the kennel doors and whiteboards in communal areas) and visually highlighting important information by colour-coding dogs as green, amber or red to reflect the level of identified risk. Discussing dogs’ needs and behaviour with the prospective new owners was also a way of communicating risk. Dog behaviour flagged during assessment was often formally discussed during scheduled staff meetings. The formal communication was further reinforced with informal conversations about dogs. These conversations dominated lunch breaks and carried on after work on social media and in other contexts. The success of risk communication in dog shelters therefore relied on formal procedures but also informal social networks.

One of the delivery companies made risk visible by recording dogs’ presence in designated books. Items for properties known to have dogs were also marked visually, to remind the delivery person about the animal when the property was approached. This was seen as particularly important as a person delivering an item may not always be familiar with the area of delivery and could be different to the person who prepared items for the delivery, increasing the reliance on written notes to communicate the location of hazards to others. However, in time-pressed environment, this procedure was not always followed. In addition, one of the Delivery Companies was trialling a standardisation system which translated the observations to a quantitative expression of risk, which was used to decide on how to manage the dog:


*“(…) One box measured the perceived severity of an individual dog attack [on a scale 1–10] and the second (…) box measured the likelihood of an attack from the same animal. (…).”*
Adam, Delivery Company.

This communication procedure was designed to represent dog risk empirically, adhering to the common definition of risk as severity multiplied by likelihood [51]. However, the severity of a bite was assessed before it occurred and depended on the subjective assessment of a dog by the delivery worker. Enactment of the communication protocol therefore required drawing on the personal experience and subjective judgement of risk. 

Failed communication: Lack of communication regarding presence or behaviour of dogs was seen as a primary reason for bites at work:


*“I think that there was a miscommunication of how aggressive this dog was. And the nurse went in to go and get the dog out of the kennel (…) that’s when the dog attacked the nurse.”*
Amy, vet nurse Dog Shelter.

In one of the Delivery Companies, employees were responsible for highlighting hazards to the managers, who updated records regarding hazards related to individual properties, evaluated hazards and decided on how it can be controlled. Managers reported however that it was sometimes difficult to get this information from frontline personnel. Frontline staff complained that occasionally they wished to report dog hazards, but were pressed for time or ignored, making them reluctant to make a report in the future. This demonstrates the role that informal social relations with colleagues play in shaping how the procedures were practiced. Schematic representation of how risk related to dog bites is communicated is shown in Figure 3. 

#### 3.2.3. Actions taken to manage perceived risk

Avoidance of the hazard: Known risks were perceived by the participants as manageable by changing human or dog behaviour. During participant-observations it was noted that one of the delivery companies had in recent years adopted a protocol which stated that employees should avoid all dogs (e.g., by crossing the street if a dog was visible, asking customers to restrain their dogs before opening the doors) and refrain from making the delivery if this was not possible. This was a departure from a previous protocol where only specific dogs identified as dangerous were avoided. The other Delivery Company advised workers to avoid dogs when believed to be dangerous.

Modifying the hazard: In dog shelters dogs could not be avoided. Attempts to modify their behaviour were made instead. During participants-observations it was observed that in some shelters, unsafe behaviours were often altered through dog training, e.g., dogs were taught to step back when a person was entering a kennel. In all shelters dog risk was also modified by changing behaviours such as taking different routes when walking around the worksite to avoid agitating dogs; using treats to distract dogs; or allowing dogs time to accustom to the veterinary procedure: 


*“If they don’t like the vets, we’ll build up things slowly. (…) We would just go into the [vet] room, feed the dog, take him back out, (…) and then eventually, you would maybe touch his ears (…). Maybe then slowly introduce a stethoscope (…), and then probably introduce the vet or the vet nurses.”*
Nick, Dog Shelter.

Matching risk with level of responsibility: In both delivery companies and shelters, responsibility could be passed to someone higher in authority when a dog hazard could not be altered. In one of the delivery companies, if a non-delivery occurred, the manager contacted the customer (paradoxically by delivering a letter to the household) explaining that the dog risk had prevented the original delivery. In shelters, dogs that were seen as more challenging were transferred to experienced staff. Managers also encouraged staff to seek the help of more experienced colleagues, demonstrating that organisational culture and leadership style can influence safe behaviour, and the role of social networks in negotiating safety.

Eliminating the hazard: The final action taken in response to perceived risk was euthanasia. Decisions regarding euthanasia followed different procedures which combined assessments of dog’s health and welfare, and judgement of risk the dog posed to themselves, employees and the public: 


*“It’s never, “Oh, it bit one person.” (…) It means this dog is really not rehomeable. It’s aggressive. Its welfare is not good. It’s a big picture. (…) We (…) [members of staff involved in dog training, day-to-day care and veterinary treatment ]have to agree. It’s a massive process.”*
Isabel, Dog Shelter.

Some shelters emphasised that dogs are only euthanised when all other options (training, medications) were exhausted and a dog’s welfare was compromised. One shelter expressed an additional concern about the negative impact on the shelter’s reputation, should a person who rehomed a dog be injured by a dog: 


*“You’ve got to think of the centre as well, because everything is now recorded. Every bite. Every bark. (…) And if that dog went out and caused trouble, there is a paper trail right back to us. And in this day and age with people suing each other, it’s just not worth the risk. You could lose your centre, your reputation (…)”*
Matt, Dog Shelter.

Barriers to following procedures: Staff in delivery companies often relied on their own methods for managing risk around dogs, such as carrying treats, despite this being not within the organisations’ procedures: 


*“We had a guy (…) who’d been bitten by a dog, but continued to interact with dogs. (…). I had to say to him, “Look, you can’t do it.” I said, “(…) You’re at risk and you put others at risk. You’re putting yourself in a situation where the [organisation] will frown on you, if you get bitten. You’re paid to deliver [items] not (…) to go stroke a dog.”*
Frank, Delivery Company.

As in the above case, a preference for a routine “*that worked for you*”, (Harriett, Delivery Company), was often given as a reason for not adhering to organisational protocols. However, not following protocols meant that one could be found responsible for having an accident and risk losing their job as it was a violation of a procedure.

Another barrier to adhering to procedures was that management of dog safety had become more bureaucratic:


*“If you go on (…) the [organisations portal], there are hundreds, if not thousands, of safety documents. Sometimes, the frontline staff, the managers, the reps don’t know where to go with it.”*
Frank, Delivery Company.

Further, adjusting behaviour did not always bring the desired results. Participants argued it was impossible to adhere to the safety procedures and deliver to all properties within the timeframe stipulated in other protocols. Another unintended consequence of following procedures and avoiding dogs was that this could upset the customers. During fieldwork, Ed described how his refusal to deliver items after a customer failed to secure the dog led to the customer’s complaint and investigation into Ed’s conduct. He was disappointed with his employer for siding with the customer and believed that the investigation undermined the value of procedures in negotiating safety. 

Staff in one of the shelters pointed out that safety procedures sometimes conflicted with other aspects of safety. For example, they were required to wear ear protectors and could not hear colleagues shouting for help. Faulty equipment was also a problem as they could not ask for help via the radio because it was outside of the range of the receiver. However, overall, compared to the delivery company, in dog shelters the experienced risks were more congruent with procedural risks and the risks identified by frontline staff were similar to those identified in protocols. A summary of actions that can be taken to managed the perceived risk of dog bites is shown in Figure 4.

#### 3.2.4. Reporting Bites and Near-Misses

Definitions of bites: Reporting of bites depended on how they were defined. One of the shelters did not have a written definition. Other shelters had written definitions of bites, where bite severity was expressed on a numerical scale which corresponded to the severity of a bite. However, in the course of fieldwork, participants did not refer to this definition, possibly because it was unknown to staff. Alternatively, in practice, staff may have assessed severity of bites with reference to own experiences. For instance, during focus groups discussions, most members of staff agreed that a bite occurs when a dog punctures skin. However, when interpreting what happened to them, they also included the context and the perceived motivation behind the incident to define a bite: 


*“When Moon got me, we did an accident instead of a bite report, because he didn’t really mean it, he just was like, “Ah,” because obviously he was in quite a lot of pain”*
Amy, Dog Shelter.

In shelters, even when near-miss reporting was possible, the perception was that bites “count against dogs” and could lead to a dog being euthanized or struggling to be rehomed. For instance, I observed Izzy assessing a large, young dog. Before the assessment started, Izzy explained that it is just a formality, as the dog is unlikely to be accepted to her kennels as he was too nervous and out of consideration for his welfare and training needs, another organisation will take care of him. Izzy described the owner as “harsh” and emphasised feeling “sorry for the dog”. During the assessment the dog lunged and bit her arm. She recorded this in the dog’s file as “uncomfortable when being handled”. Later Izzy explained that the dog did not puncture skin (a thick jacket helped) and admitted that she did not want him to have a bite “on the record”. During observations in shelters, near-misses were usually noted as dog “mouthing hard”, “playing”, “nipping”, etc., further allowing for subjective definitions of such reports. In addition to broken skin, participants in the delivery companies used dog’s breed to define the severity of a bite. For instance, in one of the delivery companies, a participant expressed his frustration when reporting a serious bite by a dog of small breed, as his colleagues did not consider it a bite.

Barriers to reporting: Similar to Izzy’s example above where she did not report the bite for fear of it ‘counting against the dog’, delivery workers discussed cases where a relationship with a customer or dog stopped them from reporting bites. In addition, when deciding whether to make a report, a number of participants assessed the potential risk to others:


*“I didn’t [report the bite]. (…) the only walk it gets in a day is at 5 o’clock in the morning, so I know it’s not a danger to the public. So I thought, “I don’t really want to be responsible for a dog getting put to sleep.””*
Georgia, Delivery Company.

In one of the Delivery Companies, a time-pressed environment and cost of incident investigations were also described as contributing to underreporting. However, the most salient factor influencing bite reporting in shelters and delivery companies was fear of being blamed. Within shelters, a bite was often seen as reflecting lack of dog-handling skills, knowledge regarding dogs’ body language or common sense: 

*“I think (…) people think “Oh, he wouldn’t do that to me,” or, “They must’ve done something wrong.” (…) there can be quite a blame game”*,Eli, Dog Shelter.

Similarly, in the delivery company frontline workers believed the main risk they faced was not that of dog bites, but the risk that emerged when they were blamed for a bite when seen not following procedures. A summary of actions taken in the course of reporting bites and near-misses is presented in Figure 5.

#### 3.2.5. Investigating bites and near-misses

In dog shelters, investigation consisted of a conversation with the victim and witnesses and reviewing notes about the dog to establish what happened. Typically, a further discussion with all staff followed to ensure awareness of any measures that may have been missed. Although the investigations aimed to learn how to prevent future incidents, one participant expressed her scepticism: 

*“[T]hey just tell you about reporting things, but nothing ever comes out of it; no lessons are learnt, or procedures changed”*,Rita, Dog Shelter.

Fear of unsettling work relationships was also quoted as a reason for not investigating bites: 


*“We didn’t go in great depths of whose fault was it, and who should have done this, [or] that. I think it’s just something that you don’t really talk about, because everyone sort of works well with each other.”*
Clair, veterinary nurse at a Dog Shelter.

In one of the Delivery Companies, the investigation process followed a form which listed all possible procedures and safety-related behaviours. It was therefore difficult not to select some boxes and conclude that an incident was blame-free. For example, one report, assigned the blame for the bite to the employer, as the “relevant equipment and supervision were not provided”. However, this bite would not be prevented through provision of relevant equipment, as the dog escaped from a property and the victim was not aware of the dog. Some participants suggested that these forms were used to assign blame for the accident, usually (but not in the case above) onto frontline staff: 


*“It’s supposed to be [thorough] investigation. In my opinion, it’s usually Inspector Clouseau who does it. (…). It’s trying to attach a blame to somebody, rather than it just being a freak accident”*
Cameron, Delivery Company.

A summary of actions taken to investigate dog bites and near-misses is presented in Figure 6.

#### 3.2.6. Learning and Teaching Safety 

The final set of procedures for managing risk around dogs was impartment of skills and knowledge that were believed to improve their safety. In all organisations training included learning organisation-specific surveillance techniques. In one of the Delivery Companies, participants were also taught the organisational procedures for reporting incidents. 

In addition, in some shelters training included different dog training techniques. Here, participants attended regular training sessions, took part in induction (also delivered in delivery companies) and mentoring programme that mixed practical and theoretical training. As well as at work, participants also learnt informally, from colleagues, family or friends: 


*“You can’t learn everything from [work training]. (…) And I suppose like, when you’re [i]n the [kennels], you do get to speak with people, and you do get to learn what they’ve been doing (…)”*
Freya, Dog Shelter.

All participants also described learning from prior incidents involving themselves or colleagues: 


*“I think definitely after you have been bitten or had an incident, it does make you a lot more mindful and it really helps you to see how a dog can progress to a certain thing.”*
Amy, Dog Shelter.

Participants described some practical barriers to learning about safety. In delivery companies, part-time staff were not always present during training; dog-safety was not always covered during inductions and peer coaching as procedures stated; and managers were not always committed or effective at delivering training. In one of the shelters, formal training was minimal due to lack of resources.

Some participants felt that learning sessions were ’tick-box exercises’ designed to show that the training took place so the organisation could not be seen as responsible for the incident due to negligence, rather than equipping staff with skills: 


*“It’s a rushed message. (…) Then, there’s a queue because, when you’ve had the message, you’ve got to sign to say you’ve had it. (…) So they sign the sheet before they’ve had the message.”*
Cameron, Delivery Company.

Learning and teaching safety was focused on identifying hazards, communicating risk and changing behaviour. It also depended on reporting and investigating bites, as what was being taught should reflect knowledge gained from previous incidents. Barriers to learning and teaching therefore undermine effectiveness of all safety procedures. A summary of the process of teaching and learning about dog safety within occupational contexts is shown in Figure 7.

## 4. Discussion

In this study, we highlighted that procedures are important in shaping perceptions of, and managing risk around, dogs. We described the procedural steps used in Delivery Companies and Dog Shelters and highlighted barriers to their implementation. Our findings raise some important points for discussion below.

### 4.1. Preventing Work Incidents vs. Preventing Dog Bites

The procedures for identifying and managing risk recognised here share similarities with those listed in oil, aviation, and nuclear industries, where hazard surveillance, effective risk communication, risk avoidance, appropriate record keeping and systems ensuring investigation and organisational learning are routine [52]. In addition, actions taken to manage perceived risk can be mapped onto the UK’s Health and Safety Risk Control Hierarchy (RCH) [53]. RCH presents risk management as steps with decreasing efficacy from risk elimination (most effective), engineering controls, administrative controls to use of personal protective equipment [53]. Accident prevention research also highlights that accidents at work occur due to a combination of: human errors; technical failures; and broader organisational factors, which encapsulate the safety culture within the organisation [54]. Our study resonates with this understanding as our participants described a combination of these factors contributing to a bite incident. Surveillance was the fundamental platform on which managing dog-related risk was built. Outside of a work context, parents of children with autism also described “vigilant parenting”- strikingly similar to the surveillance practices described by people working with dogs- as they are always ready to act, a strategy they use for identifying and managing challenging behaviours in their children [55] (p.1079). This study is different to other research into dog bite prevention, which is often based on the premise that an individual’s behaviour alone leads to dog bites and therefore advocates development of education and skills programmes (in particular for children) around hazard recognition and behaviour around dogs [56]. In fact, dog behaviour and the likelihood of a bite is shaped by multiple factors, including socialisation, genetics, context of interactions as well as behaviour of a person [57] and individual behaviour *alone* is rarely a sole cause of incidents [58]. When procedures focus primarily on changing individual behaviour, systemic changes that could improve safety are potentially ignored. 

### 4.2. Differences in Perceptions of Risk

The legal responsibility for risk management encourages tangible, measurable and thus auditable definitions of risk [59]. However, in practice, the risk of dog bites, is not experienced as an empirical phenomenon. We have highlighted how hazards and risks linked with dogs were constructed, e.g., through procedures for identifying and quantifying hazards [32]. They were also subjective: risk was identified with reference to procedures, but also individual situated knowledge, prior experiences and different contexts. For example, in dog shelters the presence of a dog was rarely seen as a hazard, but in the delivery company, it was. Perceptions of risk were also dependent on the relationship with dogs; an affinity for the dog meant participants in the shelter did not sometimes interpret a behaviour as risk. In addition, enactment of protocols and procedures required drawing on subjective judgements and assessment of risk and social relations, which also contributed to a discrepancy between “work as imagined” and “work as actually done”.

Procedures and protocols were focused on risk related to the remit of work, while in practice, individuals working with or around dogs identified other, sometimes competing risks. For instance, in the delivery companies, frontline staff identified avoiding dog bites, conflicts with colleagues and conflicts with customers as a risk, but the managers were also concerned about potential reputational risk created by suspension of a delivery. Ensuring efficiency of the delivery process was sometimes seen by the frontline staff as contributing to risk they experienced. Dog shelter staff were concerned with bite risk to themselves but also the risk to dogs when a dog’s welfare was compromised. Workers also identified the risk to the potential adopters, and associated risk to the organisation’s reputation. The complexity of how participants identified with different risks impacted on adherence to procedures. 

### 4.3. Is Safety just a Lack of Risk? 

Although safety tends to be defined as absence of risk and rarely by its own inherent qualities [54], here risk and safety have been constructed in different ways. For instance, the occurrence of dog bites did not always make frontline staff *feel* unsafe, whereas feeling blamed and fear of the incident investigation process did. Feeling safe depends on trust, often built on perception of competency and faith [60] in having the support of colleagues, managers and the organization. In this study, surveillance of risk, effective communication and feeling supported by colleagues contributed to the staff experience of safety. Conversely, lack of trust towards the organization and fear of being blamed for incidents meant that staff perceived the risk at work as high, even when organisations had procedures for risk management. 

Organisation-wide risk management protocols alone did not always make people *feel* safe. However, routines (which were often reflected in formal procedures) that “worked for you” were seen as helpful. In this way procedures, like routines, can offer a sense of security by limiting the overwhelming number of options and a need to continuously scrutinise individual behaviour for safety by providing fixed rules instead [33]. 

Participants suggested that sometimes safety procedures were used for reasons other than avoiding risk. For example, investigations of bites and the presence of safety training in one delivery company was described as a way of demonstrating that safety is maintained and a mechanism for blaming workers. It has previously been shown that organisational procedures indeed have many functions, including maintaining an organisation’s reputation [41], and helping to develop organisational memory and identity [42]. Procedures for identifying risks also created a perceived opportunity for managing those “at risk” [38] and were often interpreted as enabling employers to be able to call out workers for acting outside of organisations’ policy, regulate their behaviour and shift the responsibility for safety onto them. 

### 4.4. Study Strengths and Limitations 

Our focus was primarily on two professions: delivery work and in dog shelters (which also included veterinary work, dog training and grooming), but there are other professions where workers may also be affected, albeit less systematically (e.g., community nurses or waste collectors). Therefore, these findings may not be generalizable to other professionals who encounter dogs at work, and in particular, those who are self-employed and do not work with organisational safety protocols. 

Procedures for managing risk of dog bites are conveyed through multiple organisational policies and we cannot be sure that we reviewed all material which could potentially influence how risk is approached. The observation period in all organisations was also relatively short, limited to specific locations due to feasibility. Hence, some practices and patterns of interactions may have been missed. These limitations have been countered by using multiple methods of data collection which aided triangulation of our findings, increasing rigour. 

This research is the first to use ethnographic methods to show how people prevent bites and negotiate safety in interactions with dogs and thus offers valuable contributions to dog bite prevention. Additionally, using multiple methods of data collection (focus-group discussions, in-depth interviews, participant-observations and document analysis) improves the rigour of this study by enabling data triangulation and cross-checking of findings. Finally, although rigour of qualitative research is not related to the sample size, a large number of participants (*n* = 55) with different attitudes and histories related to dog bites took part in this research, facilitating representation of varied experiences.

### 4.5. Implications and Recommendations

Research has previously identified numerous reasons for lack of compliance with procedures, which resonate with our findings regarding safety when working around dogs. This includes an excessive number of procedures; procedures that impose heavy restrictions on workers actions; perception that compliance takes time and effort; and differences in perceptions of risk between those involved in the development of protocols and frontline staff [30]. Streamlining the number of procedures and developing procedures and definitions of dog bites with frontline staff could improve compliance.

Despite formal protocols for risk assessment, participants often drew on subjective understandings of the hazards and risks they face, which were not and could not always realistically be covered by organisations’ procedures. Safety may therefore be improved by not penalising staff for deviating from the prescribed procedures, especially if adherence to the procedures was not going to result in a safe outcome. 

Fear of being blamed had a negative impact on reporting incidents and near-misses. This is dangerous as studies from construction and aviation industries indicate that fewer near-miss reports lead to higher chances of a serious incident [37]. Addressing victim-blaming could help to improve reporting of bites so that actions can be taken to prevent them later.

## 5. Conclusions 

Procedures involving making risk visible and communicating risks are common strategies used to prevent dog bites and are the primary focus in workplace risk management regarding dogs. Discussions around dog bite prevention at work are, in fact, discussions about the role of an employer as a protector of health vis-à-vis an employee’s responsibility to protect their own health and the question of what is the extent to which an employer should protect the individual. Organisations aim to prevent bites through surveillance mechanisms, making risk visible and expecting and assisting change in worker behaviours. Some of these procedures are defined as objective processes. In practice, negotiating safety includes following procedures but also subjective, context-specific judgement of risk. The adherence with procedures depends on overlap and shared understanding of risk between organisations and individuals. Systems of bite prevention generate their own risks and the ultimate practices of staff encompass management of risks as they see, including these identified by the organisation. 

Our findings suggest that simply making and following protocols may have limited effect on managing dog bites. To avoid dog bites, management and employees need to pay attention to the contextual factors of the risks in their particular workplace, promoting safe and blame-free working climates. All in all, our study suggests that protocols may reduce risks but cannot guarantee safety. One of the effective ways to enhance safety is to build trust between employees and their organisation, between employees and their peers, and between employees and their customers.

## Figures and Tables

**Figure 1 ijerph-18-07377-f001:**
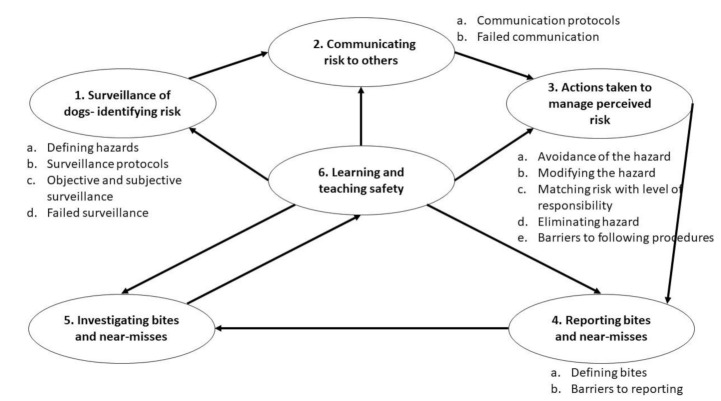
Schematic representation of the identified themes relating to dog bite prevention within work contexts. Each theme (1–6) corresponds to consecutive and co-dependent step of bite prevention. Sub-theme (a-e) are listed next to the theme.

**Figure 2 ijerph-18-07377-f002:**
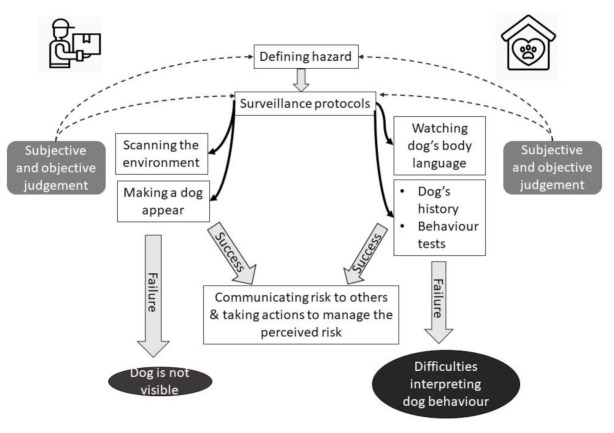
Schematic representation of the first step of dog bite prevention: surveillance of dogs: identifying the risk. In delivery companies and dog shelters, surveillance protocols depend on how the hazard is defined. The actual protocols take different forms in different organizations, however in both shelters and delivery companies, they are informed by both subjective and objective judgement. Successful implementation of surveillance protocols makes communicating risk to others possible. The protocols fail when a dog is not visible or their behaviour cannot be easily interpreted.

**Figure 3 ijerph-18-07377-f003:**
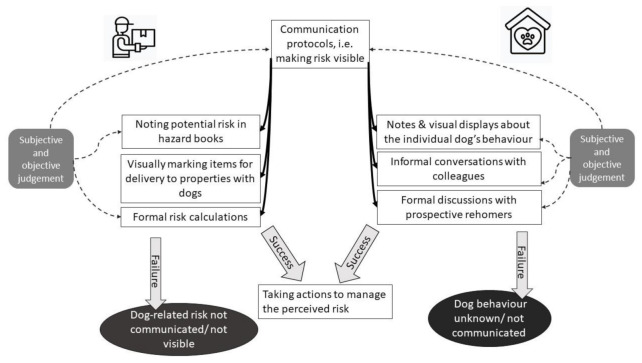
Schematic representation of the second step of dog bite prevention: communicating risk to others. In both delivery companies and dog shelters communication protocols rely on subjective and objective judgement of risk, however the specific protocols for communication differ between the organisations. Successful communication enables taking actions to manage the perceived risk. Risk is not made visible to others when it is not communicated or not known.

**Figure 4 ijerph-18-07377-f004:**
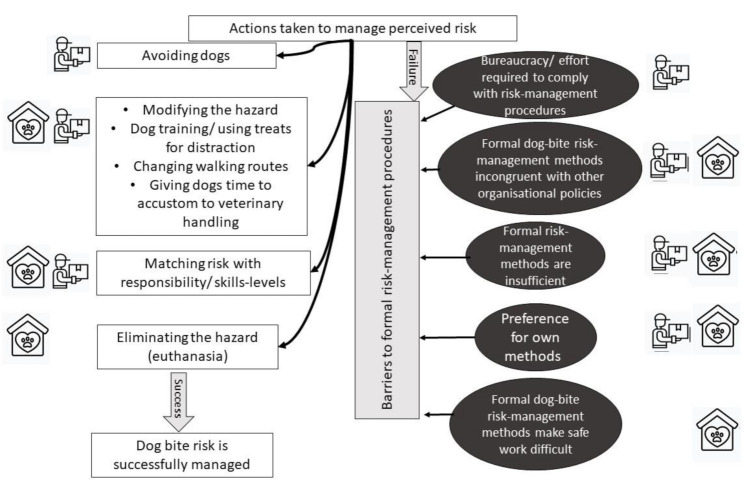
Schematic representation of actions taken to manage perceived risk. Successful implementation of avoidance, hazard modification, matching risk with responsibility/ skills-levels and hazard elimination helps to manage risk. However, a number of barriers to formal risk-management procedures in both dog shelters and delivery companies exist.

**Figure 5 ijerph-18-07377-f005:**
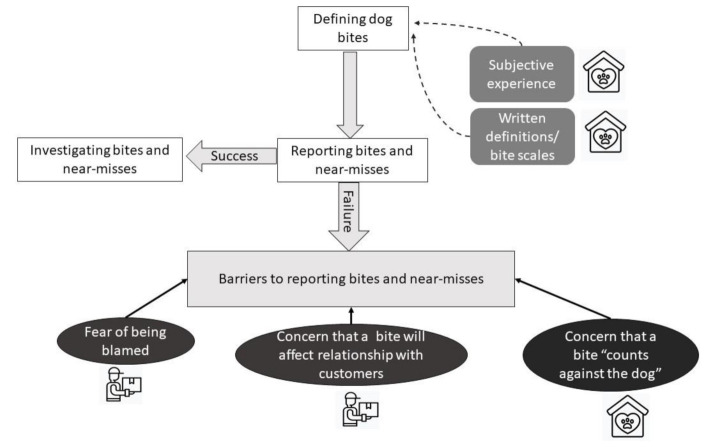
Schematic representation of reporting dog bites and near-misses. In dog shelters, actions taken to report a bite depend on bite definitions, which are informed by both subjective experiences and objective scales. In both organisations, a number of barriers to reporting dog bites has been identified.

**Figure 6 ijerph-18-07377-f006:**
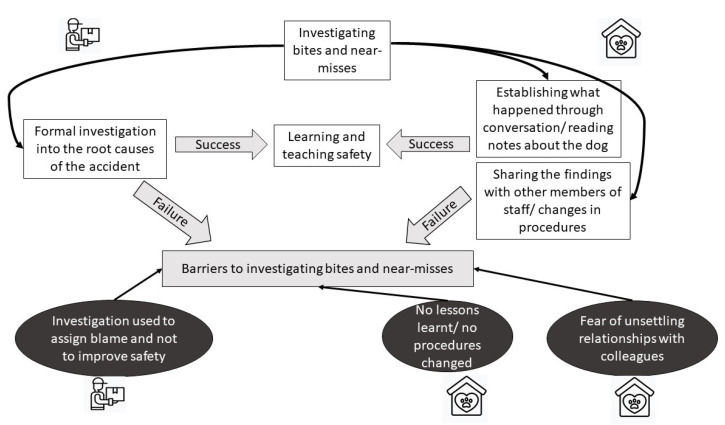
Schematic representation of actions taken to investigate dog bites and near-misses. In both organisations, a successful investigation enables learning from the incident and improvements in safety. A number of barriers to investigating dog bites has been identified in both dog shelters and delivery companies.

**Figure 7 ijerph-18-07377-f007:**
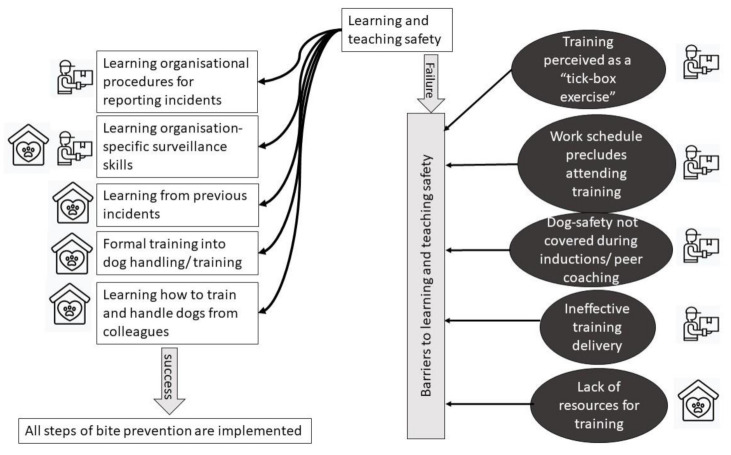
Schematic representation of learning and teaching safety. In both organisations, learning requires understanding organisaional surveillance mechanism. Multiple ways of learning have been identified. Barriers to learning/ teaching safety were observed.

**Table 1 ijerph-18-07377-t001:** Example of interview and focus group discussion questions used.

Interviews	Focus Group Discussion
How did you start working here? What, if any, are the biggest risks in your day to day work? What happened when you were bitten by a dog? How, if at all, did a bite affect your work? Is there anything that you do to stay safe around dogs? If so, could you provide an example of something that you do?	Has anyone here been bitten by a dog? Are there any dangerous aspects of your job? If so, what are they? What happened when you were bitten by a dog? How, if at all, do you prevent bites at work? How do you teach someone to be safe around dogs at work?

## Data Availability

To protect anonymity of participants, data used in this study are not publicly available. Access to the dataset can be discussed on a case-by-case basis by contacting the corresponding authors.

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
