# Peer review of "If You Don’t See the Dog, What Can You Do?” Using Procedures to Negotiate the Risk of Dog Bites in Occupational Contexts"

_ijerph, 2021, doi:10.3390/ijerph18147377_

Round 1

Reviewer 1 Report

Line 36: Estimated costs is stated as 70.8 (British Pounds). That can't be right. Was the cost 70.8 thousand or million or billion pounds. 

Line60: Add "are" 

"encounters with dogs are frequent and often unexpected"

Line 149: Instead of "in respect to" should be "with respect to"

Line 180: The percentages don't add up to 100% ( 36+63= 99). Did some participants not identify with either gender? If so, please state the reason why the % don't add to 100%

Line 623: "this research is the first to use ethnographic methods..." . Therefore it would be nice to have a paragraph added to the methods section explaining this specific methodology to the reader.

Author Response

Thank you taking time to provide us with your feedback. Below we summarised the changes implemented in this manuscript and replied to your suggestions.

Line 36: Estimated costs is stated as 70.8 (British Pounds). That can't be right. Was the cost 70.8 thousand or million or billion pounds. 

Added “millions” (line 51)

Line 60: Add "are" 

"encounters with dogs are frequent and often unexpected"

Added “are” (line 102)

Line 149: Instead of "in respect to" should be "with respect to"

Altered to with respect to (line 184)

Line 180: The percentages don't add up to 100% ( 36+63= 99). Did some participants not identify with either gender? If so, please state the reason why the % don't add to 100%

This is due to a rounding error. We’ve now added the decimal place to the percentages so that the text reads: “36.4% identified as men and 63.6% as women” (line 251)

Line 623: "this research is the first to use ethnographic methods..." . Therefore it would be nice to have a paragraph added to the methods section explaining this specific methodology to the reader.

We have now provided more detailed description of ethnographic methods (lines 152-164). The paragraph reads:

“A qualitative methodology was adopted in this study. Specifically, we used ethnographic methods which are defined broadly as “an approach to experiencing, interpreting and representing culture and society” [1, p. 18]. The purpose of ethnographic research is to identify shared patterns of ideas, beliefs, language, actions, practices, understandings in reference to the cultural context of an individual or a group and to describe them using thick and rich descriptions [2]. Ethnographic methodology does not follow any prescriptive methods, however an immersion in the culture of research participants through fieldwork and participant-observations (see section 2.2 and 2.3 for more details) is seen as important [2]. Ethnographic methods take a flexible, iterative and responsive approach to the study findings. For example, observations made during the fieldwork can be used to modify questions asked during the in-depth interviews and the preliminary interview findings can help to focus the participant-observations [2].The goal of using an ethnographic approach here was to examine people’s experiences and practices within naturalistic settings, whilst paying attention to the broader socio-cultural and environmental contexts within which these were situated [3]. To this end, a multi-sited ethnographic approach was used to explore the meaning of dog related risk at multiple fieldwork sites and in different contexts [4].”

Reviewer 2 Report

The work aims to address an unusual but exciting issue. My observations, and my suggestions, mainly concern the following points.

- In the introduction, I consider it helpful to refer, if available, to other countries besides the UK, to make the work of greater interest also for readers from other countries.

  • The results should also be streamlined with tables or other images: the paragraph is too long, and the reader's attention is easily lost.

Author Response

Thank you taking time to provide us with your feedback. Below we summarised the changes implemented in this manuscript and replied to your suggestions.

The work aims to address an unusual but exciting issue. My observations, and my suggestions, mainly concern the following points.

In the introduction, I consider it helpful to refer, if available, to other countries besides the UK, to make the work of greater interest also for readers from other countries.

This was added in lines 37-39:  “The incidence of dog bites differs between countries (e.g. 12.39 per 100,000 in Australia [5], 1.5 per 100,000 in the Netherlands [6], 25.3-30.1 per 100,000 in India [7], and 110 per 100,000 the USA[8]).”

In addition, some epidemiological data relating to the frequency of this type of injury (prevalence or incidence), even in different work context and region (if available), should be added.

We added more epidemiological data on the UK  to the information already provided (Lines 53-67). Research into epidemiology of dog bites in work-related context is scant and out dated, but we’ve added some references related to the UK, Brazil, Taiwan and the US in lines 47-61.: “Dogs are the second most commonly implicated species of animals (after insects) in all animal-related non-fatal injuries within the American workforce [9]. Among those injured, the most affected occupations are: non-farm animal caretakers, truck drivers, veterinary technicians and meter readers [9]. Moreover, in two separate studies carried out in Brazil and Taiwan, approximately 70% of surveyed postal workers reported being bitten by a dog at some point during their career [10, 11]. On average 277 people are seriously bitten each year at work in the UK, making dog bites an important issue concerning workplace safety [12]. Indeed, dog bites are the second most common cause of injury for UK postal workers (after slips and falls), with an average of 7 postal workers being bitten each working day [13]. Furthermore, 48% of 2800 surveyed Australian veterinarians were bitten by a dog during the previous 12 months [14] and in the USA 63% of surveyed veterinarians have been bitten during the course of their career [15]. In the UK, two out of three veterinarians were injured at work at some point and 78% of these injuries were animal bites, including dog bites [16]. Dog bites are also common among dog shelter workers and other professionals whose work requires entering private properties [12].”

The results should also be streamlined with tables or other images: the paragraph is too long, and the reader's attention is easily lost. Are quantitative data available? It they are, please add them.

Thank you for this suggestion. The purpose of our submitted paper is to provide a rich, qualitative understanding of how risk of dog bites is negotiated within occupational contexts rather than to quantify it.  Data suitable for quantitative analysis have already been published and we refer to this publication within this manuscript (Owczarczak-Garstecka et al. 2019; in text reference no 19). A marker of rigour in qualitative research and a crucial reporting standard (in particular in studies that rely on ethnographic methods such as this one) is provision of “thick descriptions” that help to illustrate experiences of research participants in a nuanced way. For example, defining rigour in qualitative study, Tracy (2010, p. 843) writes: “High-quality qualitative research is marked by a rich complexity of abundance—in contrast to quantitative research that is more likely appreciated for its precision”.

Provision of multiple quotes is also related with another facet of qualitative rigour, that is multivocality, defined as inclusion of multiple and varied voices in the qualitative report. Given paucity of research regarding experiences of those at risk of dog bites in the occupational contexts, we believe that the inclusion of multiple quotes is necessary. Inclusion of images taken during fieldwork is likely to compromise anonymity of research participants or their workplaces and for this reason we cannot take this step. We have however included diagrammatic representations of each of the six main themes (Figures 2-7) and hope that these improve readability.

Owczarczak-Garstecka SC, Christley R, Watkins F, Yang H, Bishop B, Westgarth C. Dog bite safety at work: An injury prevention perspective on reported occupational dog bites in the UK. Safety Science. 2019;118:595-606.

Tracy, Sarah J.2010. "Qualitative quality: Eight “big-tent” criteria for excellent qualitative research." Qualitative inquiry 16.10 : 837-851.

In the methods paragraph, the authors report “policy and training documents about preventing dog bites  were collected and included in the analysis”. It is not clear how this information was analysed. If possible, provide details that can help to understand the differences between various work contexts.

We have now clarified that all data were analysed in the same way, i.e. using thematic analysis (lines 224-228): “Interview and focus-group discussion data were transcribed ahead of the analysis. Thematic analysis was used to analyse all collected data (including policy and training documents and notes from participant-observations). This analysis included multiple steps, starting with thorough familiarisation with the collected data, followed by two coding cycles [17].”

As regard as “4.4. Study limitations”, this paragraph should be renamed “Strengths and limitations”, since there are also some strengths (lines 623-625), which in any case should be increased. In particular, authors should highlight whether the data collection method and use of the focus group are considered a strength or a weakness of the study. Please also refer to the sample size adequacy.

Thank you for this suggestion, this is a really good point. We have changed the section title to Study strengths and limitations (line 744) and the following text was added (lines 760-766): Additionally, using multiple methods of data collection (focus-group discussions, in-depth interviews, participant-observations and document analysis) improves the rigour of this study by enabling data triangulation and cross-checking of findings.

Finally, although rigour of qualitative research is not dependent on sample size (aiming for depth rather than representational breadth), a large number of participants (n=55) with different attitudes and histories related to dog bites took part in this research, facilitating representation of varied experiences.

Please check the articles, as a whole to correct some minor errors (e.g. line 52 “an objective phenomenon” instead of “an objective phenomena”).

This has been corrected (line 94) and the manuscript was checked for further errors.

Reviewer 3 Report

Dear Authors, the paper shows a certain originality for the specific goal and the proposed theme. However it seems not clear the research design. I encourage a revised form of this research theme. Please find attached some specific remarks in addition to Recommendations for Authors section.

Author Response

Thank you for taking time to provide us with suggestions on how this manuscript can be improved. Below we summarised the changes implemented in text and replied to your suggestions. We did our best to implement the required changes, however, one suggestion in the PDF document (i.e. where the text was highlighted without any comments) was unclear. We assumed that further clarification was needed in that case.

Dear Authors, the paper shows a certain originality for the specific goal and the proposed theme. However it seems not clear the research design. I encourage a revised form of this research theme. Please find attached some specific remarks in addition to Recommendations for Authors section.

Abstract:

Line 15- In a number of workplaces - it is appropriate to specify which ones

Details were added (lines 15-16): : Dog bites are a health risk in a number of workplaces, and in particular in the delivery and veterinary sectors and in dog shelters.

Line 18- Participants who encounter dogs at work were recruited- How?

Details were added (lines 18-19): Participants who encounter dogs at work were recruited using snowball sampling method.

We also added more details on participant recruitment in the text (lines 208-216):

Within each workplace, contact was first made with the organisation’s gatekeepers to facilitate the initial introductions to members of staff, advertise the study within the organisation and arrange the first interview. Later, a snowball sampling method was used to recruit participants [2]. Participants were recruited based on  their experience of bites (i.e. being bitten or avoiding bites at work), duration of employment, role within the organisation, gender and age, so that a variety of participants and contexts could be included in the study. Recruitment stopped when no new themes were emerging from analysis [17]. To ensure that the abstract remains within the Journal’s word limit, we’ve made the following additional changes:

Abstract: Dog bites are a health risk in a number of workplaces, especially in the delivery, veterinary and dog rescue sectors. This study aimed to explore how workers negotiate the risk of dog bites in daily interactions with dogs and the role of procedures in workplace safety. Participants who encounter dogs at work were recruited using snowball sampling. Ethnographic Qualitative methods (in-depth interviews, focus group discussions, and participant-observations) were used for data collection. Interview transcripts and notes  All data were coded qualitatively into themes. Six procedural steps for dog bite risk management were identified: ‘Surveillance of Dogs’; ‘Communicating risk; ‘Managing the perceived risk’; ‘Reporting bites’, ‘Investigating bites’, and; ‘Learning and Teaching Safety'. While the procedures described dog bite risk as objective, when interacting with dogs, participants drew on experiential knowledge and subjective judgment of risk. There was a discrepancy between risks that the procedures aimed to guard against and the risk participants experienced in the course of work. This which often led to disregarding procedures. Paradoxically, procedures generated risks to individual wellbeing and sometimes employment, by contributing to blaming employees for bites. Dog bite prevention could be improved by clarifying definitions of bites, involving at risk staff in procedure development, involving frontline staff in the development of procedures and avoiding blaming the victim for the incident.

Introduction: This section not provide sufficient background and not include all relevant references. See:

Hosain, M. M., M. T. Mohamed, and A. Siddiqui. "An Up to Date Guideline for Management and Prevention of Dog and Cat Bite–A Literature Review." Journal of Advances in Medicine and Medical Research (2021): 28-38.

We politely disagree regarding the relevance of this study to our work and would prefer not to include it. The Hosain et al (2001) study refers to guidelines on management of surgical management of wounds. The manuscript lacks references to the statements regarding contexts of dog bites and other facets of bite prevention. Tips for prevention included in the article are not based on research; the study also has not provided any assessment of the quality of assessed literature.

Raghavan, Malathi, et al. "Effectiveness of breed-specific legislation in decreasing dog-bite injury hospitalizations in Manitoba--what it means to researchers, policy-makers and the public." (2021).

The above is not a reference to the original research but commentary on the 2013 study entitled: "Effectiveness of breed-specific legislation in decreasing dog-bite injury hospitalizations in Manitoba”. Given that the 2021 reference is not an original research and that our manuscript refers to organisational policies and not national policies (like breed specific legislation), we politely disagree with the suggestion to include this reference.

We have however included a short description of national-level policies on dog bite prevention (lines 72-81):

“Most countries develop national policies aimed to prevent dog bites. These typically involve defining a dog bite as a punishable offence, often in conjunction with breed specific legislations (BSL) that restrict the list of dogs that can be legally owned on grounds of safety. Although studies investigating the prevalence of dog bites concluded that BSL or similar legislations was successful in Winnipeg (Canada) [18], it was not linked with a reduction in a number of dog-related hospital admissions in The Netherlands [6], Ireland [19, 20], Spain[21], Denmark [22] or England[23]. In addition, as banned breeds are usually not accurately identified [24-27], these legislations contribute to poor welfare of dogs perceived as dangerous[28-30]. Little is however known about organisation-wide policies aimed at bite prevention.”

Line 88- For example, reduction of incidents in aviation and medicine has been attributed to an introduction of checklists [11]- this reference is to generic!

Good point, this was replaced with Clay-Williams and Colligan. "Back to basics: checklists in aviation and healthcare." BMJ quality & safety 24.7 (2015): 428-431.

Lines 134-135- After individual observations, a focus group discussion was run at each of the three shelters, attracting 7, 6, and 4 participants, respectively. -  The size of groups is low.

We politely disagree. Various guidelines to focus group discussions for qualitative research suggest that the optimal group size is between 4-8 and indeed that recruiting more than 8 participants is not advisable (e.g. Krueger and Casey, 2002; Barbour and Kitzinger, 1998). In addition, our study relied on other methods of data collection, such as interviews and participants-observations.    

Krueger RA, Casey MA. Designing and conducting focus group interviews. Citeseer; 2002.

Barbour R, Kitzinger J. Developing focus group research: politics, theory and practice: Sage; 1998.

In response to comments provided by other reviewers, we added the following references and details, we hope these changes mean that the introduction section does provide sufficient background:

Lines 34-36: “The incidence of dog bites differs between countries (e.g. 12.39 per 100,000 in Australia [5], 1.5 per 100,000 in the Netherlands [6], 25.3-30.1 per 100,000 in India [7], and 110 per 100,000 the USA[8]).”

Lines 47-61: “Dogs are the second most commonly implicated species of animals (after insects) in all animal-related non-fatal injuries within the American workforce [9]. Among those injured, the most affected occupations are: non-farm animal caretakers, truck drivers, veterinary technicians and meter readers [9]. Moreover, in two separate studies carried out in Brazil and Taiwan, approximately 70% of surveyed postal workers reported being bitten by a dog at some point during their career[10, 11]. On average in the UK 277 people are seriously bitten each year at work, making dog bites an important issue concerning workplace safety [12]. Indeed, dog bites are the second most common cause of injury for UK postal workers (after slips and falls), with an average of 7 postal workers being bitten each working day [13].Furthermore, 48% of 2800 surveyed Australian veterinarians were bitten by a dog during the previous 12 months [14] and in the USA 63% of surveyed veterinarians have been bitten during the course of their career [15]. In the UK, two out of three veterinarians were injured at work at some point and 78% of these injuries were animal bites, including dog bites [16]. Dog bites are also common among dog shelter workers and other professionals whose work requires entering private properties [12].”

Materials and Methods

Lines 161-167- Thorough familiarisation with the collected data was followed by two coding cycles  [23]. During both cycles, codes were developed quasi-inductively to best summarise the content as the research objectives and theoretical literatures on risk guided researchers’ 163 focus narrowing down the coding framework [23]. During the first coding cycle, descriptive codes were developed. During the second cycle, the descriptive codes were revised 165 and analytical codes were devised. Codes were then consolidated and the main themes 166 that capture the underlying concepts were identified [23]. The first coding cycle was con-167 ducted on paper; the second cycle relied on qualitative coding software NVIVO [24].- It is necessary to describe more broadly

We have revise this part to include further details. This paragraph (lines 227-240) now reads: Interview and focus-group discussion data were transcribed ahead of the analysis. Thematic analysis was used to analyse all collected data (including policy and training documents collected in the course of fieldwork and notes from participant-observations). This analysis included multiple steps, starting with thorough familiarisation with the collected data, followed by two coding cycles [36]. During both cycles, codes were developed quasi-inductively. By this, we mean that codes were developed to best summarise the content whilst being relevant to the research objectives and theoretical literatures on risk, which guided researchers’ focus  and helped to narrow down the coding framework [36]. During the first coding cycle, descriptive codes were developed. During the second cycle, the descriptive codes were revised and replaced with analytical codes. Codes were discussed between the co-authors (S.C.O-G, C-W, F.W, R.C, and C.W) to improve rigour and consistency. Analytical codes were then grouped together on the basis of their similarity, which is how the main themes that capture the underlying concepts were identified [36]. The first coding cycle was conducted on paper; the second cycle relied on qualitative coding software NVIVO [37].

Line 310- Avoidance of the hazard (highlighted)

We assume that it is not clear that this is a sub-heading of the theme ‘Managing the perceived risk’. To clarify this, we have underlined sub-headings of all themes throughout the manuscript.

Discussion

Line 525-527- “Although details of the procedures differed due to specificity of the context of person-dog interactions, procedures took 6 forms: surveillance of dogs, making risk visible to others, taking actions to manage or avoid the perceived risk, reporting, investigating bites and near-misses and learning and teaching safety.” It is quite obvious

Thank you for pointing this out. We have revised this summary paragraph to read (lines  657-658):

In this study, we highlighted that procedures are important in shaping perceptions of, and managing risk around, dogs. We described the procedural steps used in Delivery Companies and Dog Shelters and highlighted barriers to their implementation. Our findings raise some important points for discussion below.

Round 2

Reviewer 3 Report

Dear Authors,

thank you for accepting my suggestions. The quality of this work has certainly improved. I have no other comments on this.

Kind regards